# PT-Symmetric *LC* Passive Wireless Sensing

**DOI:** 10.3390/s23115191

**Published:** 2023-05-30

**Authors:** Dong-Yan Chen, Lei Dong, Qing-An Huang

**Affiliations:** Key Laboratory of MEMS of the Ministry of Education, Southeast University, Nanjing 210096, China; 230228385@seu.edu.cn

**Keywords:** exceptional point, *LC* passive wireless sensor, non-Hermitian Hamiltonians, parity–time symmetry

## Abstract

Parity–time (PT) symmetry challenges the long-held theoretical basis that only Hermitian operators correspond to observable phenomena in quantum mechanics. Non-Hermitian Hamiltonians satisfying PT symmetry also have a real-valued energy spectrum. In the field of inductor–capacitor (*LC*) passive wireless sensors, PT symmetry is mainly used for improving performance in terms of multi-parameter sensing, ultrahigh sensitivity, and longer interrogation distance. For example, the proposal of both higher-order PT symmetry and divergent exceptional points can utilize a more drastic bifurcation process around exceptional points (EPs) to accomplish a significantly higher sensitivity and spectral resolution. However, there are still many controversies regarding the inevitable noise and actual precision of the EP sensors. In this review, we systematically present the research status of PT-symmetric *LC* sensors in three working areas: exact phase, exceptional point, and broken phase, demonstrating the advantages of non-Hermitian sensing concerning classical *LC* sensing principles.

## 1. Introduction

Inductor–capacitor (*LC*) passive wireless sensors have the key characteristics of minimum volume and theoretically infinite life. These sensors offer significant advantages for measuring critical parameters in harsh and remote environments, such as confined spaces and mechanical rotating structures, where wired connections are difficult and challenging. Therefore, there is extensive attention being paid to the development and innovation of wireless passive sensors.

First proposed by Collins in 1967 [1], the *LC* passive wireless sensor is usually composed of an inductor and a sensitive capacitor, whose resonant frequency is modulated by parameters of interest. An external impedance analyzer interrogates the sensors via near-field coupling to realize the purpose of passive wireless detection. Due to their low power consumption, low operating frequency, low cost, good long-term stability, and remote query capability [2], *LC* passive wireless sensors have become a research hotspot in the Internet of Things (IoT) era. They have been widely used in biomedical treatment [3,4,5,6], industrial monitoring [7,8,9,10], aerospace [11,12,13], wearable devices [14,15], and implantable fields [3,16].

However, the small size of the *LC* sensor limits their readout distance due to the dispersion of the magnetic fields and the weak magnetic coupling between the sensor and readout coil. Several methods have been proposed to prolong the readout distance, including increasing the size of sensor inductors, which does not meet the requirements of miniaturization for sensors in the IoT era [17]. Some scholars have also proposed that the use of a ferrite magnetic core instead of the air magnetic core can improve the measurement range [18]. However, the ferrite material with constant permeability must be chosen for the coupling magnetic field magnitudes and resonant frequencies. Inspired by the principle of strongly coupled magnetic resonances [19], it is found that adding a repeater between the readout coil and *LC* sensor to enhance the magnetic coupling can also realize telemetry over a longer distance [20,21]. However, this results in multiple peak frequencies and sensitivity decay. The cyclic-scanning repeater [22] and adaptive repeater [23] have been put forward, not only increasing the readout distance but improving the difficulty of sensitivity degradation. In addition, impedance matching can also effectively improve the signal strength and readout distance of the *LC* passive wireless sensor system [24]. However, for the change in the sensor’s sensitive capacitance in practical applications, impedance matching needs to adjust multiple parameters in the readout circuit at the same time, which is extremely tedious.

With the advancements in quantum mechanics, it has been discovered that the PT-symmetry-breaking regime can be utilized to extend the remote distance of the *LC* passive wireless sensors [25]. The experimental results indicate that the combination of *LC* passive wireless sensors with PT symmetry can provide a four-fold increase in readout distance while keeping the same level of sensitivity. This innovative design strategy offers promising possibilities for *LC* passive wireless sensors.

## 2. PT Symmetry

In 1998, Bender and Boettcher proposed the PT symmetry scheme in quantum mechanics [26]. They argued that non-Hermitian Hamiltonians satisfying PT symmetry have a real-valued energy spectrum, which is quite different from the theoretical basis held for a long time that only Hermitian operators correspond to the observable phenomena in quantum mechanics. Therefore, the proposal of non-Hermitian Hamiltonians revolutionarily extended quantum theories into new regimes. However, achieving a balanced distribution of loss and gain required for PT symmetry is challenging from an experimental perspective, especially at higher operating frequencies [27]. Initially, no actual case satisfied PT symmetry on the micro level until the discovery of a mathematical equivalence between the single-particle Schrodinger equation and the Helmholtz wave equation. Since then, PT symmetry quantum mechanics have garnered widespread attention as it offers new design strategies for devices with novel functions. In 2009, a practical system satisfying PT symmetry was realized by Guo et al. through analogizing the Schrodinger equation with the macroscopic optical wave equation [28], which opened the door to PT-symmetric optics and photonics [29,30]. PT symmetry was achieved in acoustics in 2014 by analogizing the Schrodinger equation with the wave equation of sound [31]. From 2017 onwards, the introduction of PT symmetry in *LRC* resonators has been found to achieve the enhancement of sensor performance, leading to further exploration of PT symmetry in the field of electronics.

Nowadays, PT-symmetric quantum mechanics has been extensively studied in various fields such as photonics, optomechanics [32,33], photonic crystals [34,35], acoustics, electronics, fully integrated electronics [36], and metamaterials [37]. Due to the one-to-one correspondence between the anti-PT symmetry and PT symmetry, APT-symmetric systems have also attracted widespread interest in generating novel non-Hermitian systems and devices [38]. Intriguing phenomena can be achieved, such as coherent perfect laser absorbers [39], optical isolators, optical circulators [40,41], and unidirectional acoustic cloaks realizing the unidirectional transparency of media [31], etc. In the field of electronics, PT symmetry is mainly used to achieve wireless power transmission [42] and improve the performance of *LC* passive wireless sensors, such as multi-parameter, sensitivity, and readout distance.

### 2.1. PT Symmetry in Quantum Mechanics

In quantum mechanics, *P* is defined as a parity inversion operator, representing the mirror image of continuous space coordinates from x to −x. The parity operator has effects x→−x,p→−p,i→i, which is a linear operator. *T* is the time reversal operator, implemented by the transformation from *t* to −t. Under the inversion of *T,* there is x→x,p→−p,i→−i, so the *T* operator is anti-linear. For the parity–time operator *PT,* there will be x→−x,p→p,i→−i, which is also an anti-linear operator. If the Hamiltonian *H* of the system and *P* operator satisfy $\left[H,P\right]=0$, the system satisfies parity inversion symmetry. In the same way, when the Hamiltonian *H* and *T* operators satisfy $\left[H,T\right]=0$, the system will satisfy time inversion symmetry. The PT-symmetric system is not symmetric under *P* or *T* operator separately but satisfies space-time reflection symmetry, whose Hamiltonian *H,* together with the parity–time combined operator *PT,* satisfy $\left[H,PT\right]=0$. The key point of PT-symmetric quantum mechanics is that the eigenvalues of a system can be real, even if the Hamiltonian *H* of the system is non-Hermitian. However, it should be noted that only when the Hamiltonian *H* and eigenfunction *φ_n_*(*x*) of the system satisfy PT symmetry at the same time can the system possess a real-valued energy spectrum. That is to say, the following two conditions should be met [26]:(1)$$\left[H,PT\right]=HPT-PTH=0,PT\varphi_{n}(x)=\lambda_{n}\varphi_{n}(x).$$

The Hamiltonian of the system satisfies the eigenequation:(2)$$H\varphi_{n}\left(x\right)=E_{n}\varphi_{n}\left(x\right).$$

Therefore, it can be concluded that
(3)$$H\varphi_{n}(x)=E_{n}^{*}\varphi_{n}(x).$$

Considering the uniqueness of the eigenenergy spectrum, we can see that $E_{n}^{*}=E_{n}$. The eigenvalues of the system are still real, and the system retains PT symmetry. In the opposite case, the eigenstates of the system do not satisfy PT symmetry: $PT\varphi_{n}(x)\neq \lambda_{n}\varphi_{n}(x)$. Similarly, it will be found that the energy spectrum of the system will appear in the form of complex conjugate pairs, and the PT symmetry of the system will be destroyed. Bender introduced a Hamilton operator *H* to verify the properties of PT-symmetric systems [43]:(4)$$H=\left[\begin{array}{cc} re^{j\theta } & \mu \\ \mu & re^{-j\theta } \end{array}\right],$$
where *r*, *μ*, and *θ* are real numbers. The Hamiltonian *H* of Equation (4) is non-Hermitian, but satisfies PT symmetry. Parity operator *P* can be expressed by a matrix as follows [43]:(5)$$P=\left[\begin{array}{cc} 0 & 1\\ 1 & 0 \end{array}\right]$$

One can derive the following equation:(6)$$PTHPT=\left[\begin{array}{cc} 0 & 1\\ 1 & 0 \end{array}\right]\left[\begin{array}{cc} re^{-j\theta } & \mu \\ \mu & re^{+j\theta } \end{array}\right]\left[\begin{array}{cc} 0 & 1\\ 1 & 0 \end{array}\right]=\left[\begin{array}{cc} re^{j\theta } & \mu \\ \mu & re^{-j\theta } \end{array}\right]=H$$

This verifies again that the Hamiltonian *H* satisfies PT symmetry. Compared with the coupled mode equation, the PT-symmetric Hamilton operator can be transformed as follows: ω=rcosθ;γ=rsinθ. Then, the *H* can be expressed as follows:(7)$$H=\left[\begin{array}{cc} \omega +j\gamma & \mu \\ \mu & \omega -j\gamma \end{array}\right]$$

Solve the eigenvalue equation as follows:(8)$$E_{\pm }=\omega \pm {\left(\mu^{2}-\gamma^{2}\right)}^{1/2}$$

The eigenvalues are related to the coupling coefficient *μ* and the loss factor *γ*. Figure 1 shows a schematic transition of real and imaginary parts in the eigenvalues from exact PT symmetry to broken PT symmetry.$\mu >\gamma ,E_{n}=\omega \pm {\left(\mu^{2}-\gamma^{2}\right)}^{1/2}$. The eigenvalues are real-valued numbers, and the system is in the PT-symmetric exact phase;μ=γ, the eigenvalues will merge. This point is described as the exceptional point (EP), at which the eigenstates are also merged;$\mu <\gamma ,E_{n}=\omega \pm j{\left(\mu^{2}-\gamma^{2}\right)}^{1/2}$. The eigenvalue is a complex conjugate, and the system is in the PT-symmetric broken phase.

PT symmetry introduces an exciting phenomenon: phase transition, which arises from spontaneous breaking around EPs. This kind of transition unveils its non-Hermitian nature, where the eigenvalue spectrum of systems changes from real value to complex value. Therefore, the appearance of PT symmetry does not conflict with conventional quantum mechanics but is rather a generalization in the complex domain [26,43,44].

### 2.2. PT Symmetry in LC Sensors

According to Kirchhoff’s law, all the spatial symmetries can be reduced to the network topology problem in the circuit. As long as the node topology of the network is satisfied and selects connection elements appropriately, the real physical spatial symmetry will not need to be considered, which greatly facilitates the implementation of PT symmetry in the circuit [45]. Similarly, the *P* operation is equivalent to swapping the subscripts of the corresponding pair of circuit elements. Additionally, the time inversion operator *T* is equivalent to changing the sign of the pure resistive impedance element, while the reactance part remains the same. Therefore, the negative resistance needs to be introduced, which is essentially equivalent to an amplifier [46].

Figure 2 shows the basic structure of the PT-symmetric *LC* passive wireless sensor, a pair of inductively coupled *LCR* resonators. The sensor has a positive resistance ($R_{s})$ as a loss end, and the reader has a negative resistance $\left(R_{r}\right)$ as a gain end with energy amplification. To satisfy PT symmetry, the components in the loop are required to meet the following requirements: $L_{r}=L_{s}=L,C_{r}=C_{s}=C,{-R}_{r}=R_{s}=R$. The mutual inductance between the two inductors is $M=k\sqrt{L_{r}L_{s}}=kL$, where *k* (0<k<1) is the coupling coefficient between inductors.

According to Kirchhoff’s law,
(9)$$\frac{i_{r}}{j\omega C}+j\omega Li_{r}-{Ri}_{r}+j\omega Mi_{s}=0,$$
(10)$$\frac{i_{s}}{j\omega C}+j\omega Li_{s}+{Ri}_{s}+j\omega Mi_{r}=0,$$
the following matrix equation can be obtained:(11)1jωC+jωL−RjωMjωM1jωC+jωL+Riris=0.

Make the following substitution
(12)$$\omega_{0}=\frac{1}{\sqrt{LC}},\;\;\gamma_{r,s}=R_{r,s}\sqrt{\frac{C}{L}},$$
where $\omega_{0}$ is the natural frequency at both ends of the system, ω is the angular frequency of the system, and γ is the gain or loss parameter; the following matrix is obtained after sorting:(13)ω02−ω2−jωRL−kω2kω2ω02−ω2+jωRLiris=0.

The eigenvalues of the system can be rewritten as
(14)$$\omega_{n}=\pm \sqrt{\frac{2-\gamma^{2}\pm \sqrt{4k^{2}-4\gamma^{2}+\gamma^{4}}}{2\left(1-k^{2}\right)}\omega_{0}}.$$

When $\gamma <\gamma_{EP}$,
(15)$$\gamma_{EP}=\sqrt{2}\sqrt{1-\sqrt{\left(1-k^{2}\right)}},$$
The system is in the PT-symmetric exact phase and has two unequal positive real-valued eigensolutions:(16)$$\omega_{1,2}=\frac{\sqrt{-2+\gamma^{2}\mp \sqrt{4k^{2}-4\gamma^{2}+\gamma^{4}}}}{\sqrt{2\left(-1+k^{2}\right)}}\omega_{0}.$$

The values of capacitance and resistance can be deduced from the above two equations:(17)$$C=\frac{1}{L\sqrt{1-k^{2}}\omega_{1}\omega_{2}},$$
(18)$$R=L\sqrt{{(\omega }_{1}^{2}+{\omega_{2}^{2})(k}^{2}-1)+2\omega_{1}\omega_{2}\sqrt{1-k^{2}}}.$$

Therefore, we can extract the two resonance frequencies to judge the change in system parameters, realizing the purpose of passive wireless detection ultimately. The state of the PT-symmetric *LC* system depends on the values of $\gamma \;\mathrm{and}\;\gamma_{EP}$ (15).$\gamma <\gamma_{EP}$, The eigenvalues $\omega_{1,2}$ are real-valued numbers, and the system is in the PT-symmetric exact phase. The whole system will maintain equilibrium, and the total energy is also conserved;$\gamma =\gamma_{EP}$, this special point is the EP point. The system is still in the PT-symmetric phase, but the eigenvalues and eigenstates coalesce. In the following sections, we will see that the abrupt nature of phase transition results in intriguing phenomena;$\gamma >\gamma_{EP}$, The eigenvalues $\omega_{1,2}$ are a pair of conjugate complex numbers, and the system is in the PT-symmetric broken phase. In this region, the whole system is not in an equilibrium state, and the total energy is not conserved, which is exactly the opposite of the exact phase.

*LC* passive wireless sensors will combine with the PT-symmetric quantum theories to achieve improvement and changes in the sensors’ performance, respectively, in the exact phase, EP point, and broken phase.

## 3. PT-Symmetric *LC* Sensing Systems

### 3.1. PT-Symmetric LC Sensors in Exact Phase

Superior detection and great robustness to noise have been a long-sought goal for *LC* microsensors, which require sensors to exhibit high Q-factor and sharp, narrowband resonant reflection dips. However, due to the inevitable power dissipation caused by skin effects and eddy currents [47], traditional *LC* microsensors usually have a low modal Q-factor [48]. A PT-symmetric telemetric sensor system designed to work in the exact PT-symmetric phase can exhibit sharp and deep resonant reflection dips, boosting the effective Q-factor and sensitivity. Moreover, the single-port scattering multi-frequency resonance characteristics in the exact PT-symmetric phase can realize the simultaneous measurement of multiple parameters of the *LC* single-resonance circuit sensor.

#### 3.1.1. High Q-Factor and Deep Reflection Dips

Sakhdari et al. presented an *LC* wireless readout sensor based on the theory of PT symmetry [49]. It is pointed out that the exact PT-symmetric phase results in real eigenfrequencies, which introduces the narrow band and sharp-peaked resonances. Compared with the conventional coil-antenna readout technique, it is evidently seen that the PT-symmetric wireless readout sensors enable the exhibition of much sharper reflection dips and greater Q-factor when operated in the exact phase. They explained the cause of improvement to be that the reflectionless property in one-port measurement owes to the impedance matching. In the exact PT-symmetric phase, the input impedance looking into the active reader can be matched to the impedance of the generator (*Z*_0_) at the eigenfrequencies, leading to the dips observed in the reflection spectrum.

Based on the second-order PT-symmetric circuit, Yin et al. [50] proposed a sandwich-type wireless capacitance readout mechanism based on a perturbed PT-symmetric electronic trimer consisting of a gain–neutral–loss *LC* resonator chain to realize a higher Q-factor. Different from the standard second-order and third-order PT-symmetric systems, the proposed sandwich-type sensing system can maintain a high Q-factor in the whole range of *k*, rather than only in the exact phase, and extend the interrogation distance while maintaining ultrahigh resolution even in the weak coupling regime.

#### 3.1.2. The Multi-Parameter Sensitive Measurement

In the practical application of wireless sensor networks, it is necessary to measure multiple target parameters in the environment simultaneously. Therefore, inspired by the method of Ren et al. [51], Zhou et al. utilized the multi-frequency resonance characteristics of the PT-symmetric *LC* passive wireless sensor system in the exact PT-symmetric phase to realize the simultaneous measurement of multiple parameters in the single-port *LC* sensor [52]. Figure 3 shows the equivalent circuit model for single-port measurement of a PT-symmetric system, where *Z*_0_ is the impedance of the frequency network analyzer. $\omega_{1,2}$ can be derived as
(19)$$\omega_{1,2}=\frac{\sqrt{2L+{CR}^{2}\mp \sqrt{C^{2}R^{4}-4LCR^{2}+4L^{2}k^{2}}}}{\sqrt{2\left(1-k^{2}\right)L^{2}C}}\omega_{0}.$$

They analyzed the two frequencies $\omega_{1,2}$ by extracting the amplitude extremum of the amplitude-frequency characteristics curve of *S*_11_ in the PT-symmetric exact phase, and obtained two sensitive parameters by decoupling the equations. This method improves the limitation of the double-parameter scheme that requires known coupling coefficients. In addition, a three-parameter sensing method for a single-loop *LC* sensor is proposed by using the three resonant frequencies at the zero phase of amplitude-frequency characteristics, which provides a feasible way to realize *LC* passive wireless multi-parameter sensing.

#### 3.1.3. Generalized Parity–Time Symmetry

Despite the advantages of traditional PT-symmetric systems, practical implementations of an exact PT-symmetric phase for the *LC* wireless sensors still encounters many difficulties. For instance, given the limited physical space of medical bioimplants and MEMS sensors, the inductance of the sensor’s microcoil is usually designed to be smaller than that of the reader’s coil. Although downscaling the reader’s coil does not affect the match between them, it will reduce the mutually inductive coupling and degrade the operation of the wireless sensor.

To overcome the limitations in the spatial dimensions and significantly improve the performance of PT-symmetric *LC* sensors, Chen et al. introduced the concept of parity–time-reciprocal scaling symmetry (PTX), consisting of an active reader (−*RLC* tank), wirelessly interrogating a passive microsensor (*RLC* tank) [48]. In the system, *x* is the reciprocal-scaling coefficient, an arbitrary positive real number, which means that PTX-symmetric systems do not have to maintain a strict symmetry mechanism as PT-symmetric systems do. Despite the introduction of the *X* operator, the whole system has an unequal gain and loss coefficient; the PTX-symmetric system and its associated PT-symmetric system exhibit exactly the same eigenspectrum and bifurcation points, thus leading to sharp and deep resonant reflection dips. The scaling operation *X* offers an additional degree of flexibility in system designs, allowing arbitrary scaling of the coil inductance and other parameters. More importantly, the scaling provided by the *X*(*x >* 1) operator leads to linewidth sharpening, boosting the Q-factor, sensing resolution, and overall sensitivity.

### 3.2. PT-Symmetric LC Sensors Based on Exceptional Points (EPs)

As we already described, the PT-symmetric EP is the merging point in the physical system, at which the eigenvalues and eigenvectors merge and the eigenfrequencies undergo a bifurcation process branching out in the complex plane. It is precisely due to this large bifurcation effect that the PT-symmetric system can significantly exhibit greater sensitivity and resonance frequency shift when operated around EP.

#### 3.2.1. PT-Symmetric *LC* Sensors with Enhanced Sensitivity

In 2019, an ultrasensitive wireless displacement sensing technique operated around EP was proposed [53]. Specifically, such a non-Hermitian electronic system obeying the PT symmetry achieves drastic frequency responses and high sensitivity, well beyond the limit of conventional fully passive wireless displacement sensors. They theoretically point out that, for the EP case, the normalized resonance frequency shift can be derived as $∆\omega /\omega_{0}\approx \sqrt{c∆k}$, where *c* is a constant associated with the *γ*. However, the resonance frequency is approximately given by $∆\omega /\omega_{0}\approx 2c∆k$ in the conventional passive-coil reader. Obviously, the smaller the ∆k is, the more excellent the sensitivity of EP sensors is. The PT-symmetric system can achieve more excellent sensitivity.

The same year, Dong et al. demonstrated that a reconfigurable wireless system subcutaneously implanted in a rat could automatically lock to an EP, showing the spectral response of $∆\omega \approx k^{2/3}$, beyond the ∆k linear limit encountered by existing readout schemes [54]. The experiment proves the minimum coupling coefficient k that can produce detectable frequency shifts lowered from 0.037 to 1.4 × 10^−3^, or about 26 times, which provides a new idea for extending the detection distance.

Based on the presence of the gain and the loss, Zhou et al. established a second-order PT-symmetric *LC* passive wireless sensor to analyze the frequency responses of asymmetric and symmetric perturbations shown in Figure 4 [55]. Although the results show that the frequency splitting caused by the two kinds of perturbations is proportional to the square root of the perturbation, the sensitivity of the asymmetric perturbation is the highest. Theoretical derivation shows that the asymmetric perturbation is more sensitive than $1/\sqrt{2\gamma }$ times the symmetric.

#### 3.2.2. *LC* Sensors Based on High-Order PT-Symmetric EP and DEP

Although sensors working in the second-order EP have demonstrated sensitivity responses beyond conventional sensors, the emergence of the higher-order PT symmetry theory still opened the door to a new *LC* sensing world. The eigenfrequency splitting of a higher-order PT symmetry telemetry wireless *LC* sensor will be more dramatic. An *N*-level PT-symmetric system can obtain *N + 1* eigenfrequency branches, leading to the ever-boosted level of eigenfrequency bifurcation and greater sensitivity to perturbation [56]. In addition, increasing the order *N* can also downshift the exceptional point so that the exact symmetry phase can be obtained with real eigenfrequencies and high Q-factor resonances. Figure 5 illustrates *N*th-order PT-symmetric *RLC* telemetric electric circuit with *N* = 2 and *N* = 3.

Inspired by the higher-order PT symmetry, a battery-free, wearable wireless sensor based on the third-order was presented, which is capable of simultaneously detecting temperature and relative humidity during the wound healing process with extraordinary sensitivity and accuracy [57]. Similarly to the schematic of third-order PT in Figure 5, the sensing system also consists of a passive RLC tank, a neutral LC tank, and an active −RLC tank inductively coupled with the adjacent one. In particular, the passive RLC tank is composed of a thermistor and a capacitive humidity sensor for responding to changes in temperature and humidity simultaneously. During the experimental demonstration, the resonant frequencies appear as dips in the reflection spectra, which respond sensitively and differently to changes in temperature and relative humidity.

Based on further studies on higher-order PT symmetry, in 2019, the concept of divergent exceptional points (DEPs) was proposed and indirectly realized in third-order PT-symmetric electronic circuits composed of inductively coupled −*RLC*, *LC*, and *RLC* oscillators [58]. DEPs are the combination of EPs with a mathematical divergent singularity. When the inductive coupling strength is close enough to DEPs, the real eigenfrequencies of the system will rapidly diverge in response to perturbation, which may be more significant than the regular bifurcation process of EPs, providing more ultrahigh sensitivity and resolvability.

Sakhdari et al. discussed the higher-order electronic circuitry again in 2022 [59]. Although the critical inductive coupling strength necessary for DEPs decreases with increasing the order of PT electronic system, some practical problems are unavoidable in a very high-order electronic circuitry, such as noise and interference, unreliability, and instability during measurement. As a result, there exists a subtle compromise among various aspects of the sensing system, including the degree of inductive coupling strength, sensitivity, stability, and spectral noises. That is the reason why the higher-order theory has not been applied widely and practically so far.

#### 3.2.3. Noise and Precision of *LC* Sensors Based on the EPs

Although numerous theories and experiments have confirmed that the PT-symmetric EP sensor does achieve enhanced sensitivity, Langbein et al. pointed out back in 2018 that the sensitivity was a very ambiguous quantity [60]. It can either refer to the sensors’ transduction coefficient from the quantity to be measured to some intermediate output quantity, or to the smallest measurable change in the input quantity given by the noise of the output. The second expression is also defined as the precision of the measurement, different from the sensitivity in the traditional sense. Due to the existence of additional noise during the gain process in signal, EP has higher noise in measuring the perturbation than DP. Therefore, it is obvious that the complex frequency splitting of the EP sensor is not suitable for estimating the precision.

Similar views on the relationship between sensitivity and noise were proposed by Duggan et al. [61]. They thought that noise could stem from a variety of sources, including mechanical vibrations, thermal noise, quantum uncertainty, or fundamental resolution limits, and never be fully eliminated. Once the output noise is exactly enhanced by the same level as the enhanced sensitivity, there will be no enhancement of precision. With a thorough discussion of the inevitable noise and enhanced precision in EP sensors, the doubts about the actual sensitivity of sensors operating near an EP have been explained to some extent.

#### 3.2.4. Solutions to Overcome Noise Effect

Considering the above issues, a PT-symmetric sensing circuit bearing a sixth-order EP is fabricated to address the potential drawbacks of EP sensing, including both fundamental resolution limit and noise effects [62]. The sixth-order EP achieved an enhanced resonance shift proportional to the fourth-order root of the perturbation strength and enhanced sensitivity compared to corresponding DP sensors. At the same time, due to the low-pass feature of the sixth-order circuit by choosing a proper working capacitance in the resonator, thermal noise is mitigated down to the identical level of the DP sensing scheme, solving the problem of noise limitation.

Kononchuk et al. proposed a PT-symmetric electromechanical accelerometer-based EP in the proximity of the detuning from a transmission peak degeneracy (TPD) as a measurement of the sensitivity to the influence of excess noise effects [63]. TPD would form when the system is weakly coupled to transmission lines. Thanks to the existence of coupling *C_e_* shown in Figure 6, the TPDs occur at distinct parameter values from the EP, ensuring completeness of the eigenbasis. The experiments demonstrated a three times signal-to-noise ratio enhancement and a ten times increase in responsivity to small perturbations compared to configurations operating away from the TPD.

### 3.3. PT-Symmetric LC Sensors in Broken Phase

Zhou et al. proposed and demonstrated that the PT symmetry-breaking regime could be utilized to extend the remote distance of the *LC* passive wireless sensors [25]. The impedance matching enables non-reflection energy transfer, and the negative resistance (−*R*_0_) at the reader has an amplifying effect on the signal. The combination of impedance matching and negative resistance realizes longer read-out distances compared to conventional sensors. Experimental results show that the read-out distance of PT-symmetric sensors is approximately four times as long as that of conventional sensors, while keeping the same sensitivity. The schematics of conventional and PT-symmetric sensing systems are shown in Figure 7a, in which *Z*_0_ is the characteristic impedance of the transmission line of the Vector Network Analyzer (VNA). As previously mentioned, the eigenfrequencies are coming in complex conjugate pairs with nonvanishing real parts in the PT-symmetric broken phase. Different from the double-frequency resonance of the exact phase [52], when the system enters the PT-symmetric broken phase, $\omega_{1,2}$ will be a complex conjugate pair. The real-domain sweep cannot read the imaginary information, so the single frequency at the minimum value of the reflection coefficient *S*_11_ is the real part of its eigenfrequency $\omega_{r},$
(20)$$\omega_{r}=\frac{1}{2}\sqrt{\frac{2-\gamma^{2}+2\sqrt{1-k^{2}}}{1-k^{2}}}.$$

Figure 7b shows the change in *S*_11_ at the resonant frequency with the coupling coefficient *k*, intuitively illustrating the enhancement effect of the readout distance. Even in the weak coupling regime of the PT-symmetric broken phase corresponding to a longer distance, the reflection coefficient almost remains constant as the coupling strength weakens. In contrast, the conventional method can only obtain the dip of the *S*_11_ signal over a shorter distance. Therefore, longer detection distances can be obtained by working in the broken phase of PT-symmetric *LC* passive wireless sensing systems, which overcomes the long-standing challenge of extending the interrogation distance of the scaled-down *LC* passive wireless sensor.

Moreover, the third-order PT-symmetry-breaking regime also shows the same superiority in long interrogation distances [50]. A sandwich-type wireless capacitance readout system consisting of a gain–neutral–loss *LC* resonator chain is fabricated, extending the interrogation distance in the PT-symmetric broken phase compared to the standard second-order PT-symmetric system. PT symmetry-breaking regimes for the *LC* passive wireless sensors can be used in the interrogation of implanted or sealed fields, where a longer interrogation distance is necessary.

Although there are still many controversies regarding the performance of the PT-symmetric sensors, the PT symmetry LC passive wireless sensors have been widely applied and developed in many fields, including biomedical and biotelemetry applications, industrial measurement and telemetry, and various wearable and implantable devices, as listed in Table 1.

## 4. Conclusions

This review systematically summarizes the research status of PT-symmetric *LC* sensors in the exact phase, exceptional point, and broken phase, demonstrating the advantages and controversies of non-Hermitian sensing with respect to classical sensing principles. However, there are still many controversies regarding the inevitable noise and actual sensitivity of the EP sensors. The innovative construction strategy of the PT sensors still needs to be further studied and explored to avoid the negative effect of noise.

In general, numerous works have demonstrated the real superiority of PT-symmetric *LC* sensing systems, including higher Q-factor and sensitivity, longer interrogation distance, and multi-parameter sensing method, which will shine brilliantly in the future of *LC* sensing systems.

## Figures and Tables

**Figure 1 sensors-23-05191-f001:**
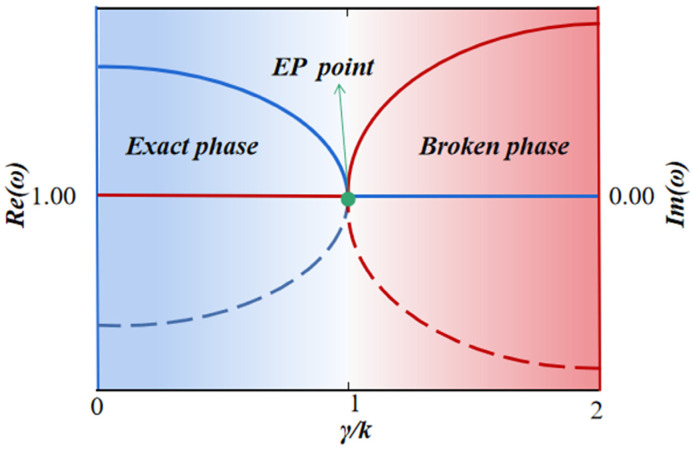
The schematic transition of real and imaginary parts in the eigenvalues from PT-symmetric exact phase to PT-symmetric broken phase.

**Figure 2 sensors-23-05191-f002:**
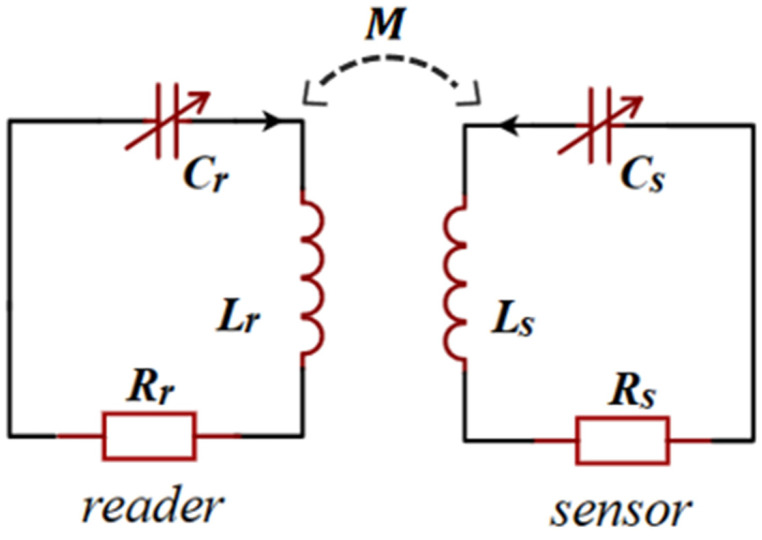
The schematic of the PT-symmetric *LC* passive wireless sensor.

**Figure 3 sensors-23-05191-f003:**
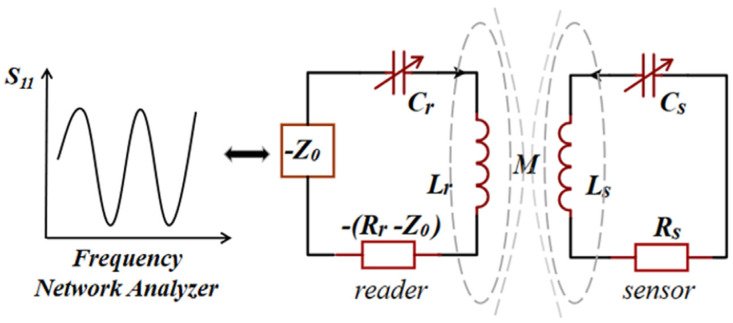
The schematic of PT-symmetric *LC* passive wireless sensor for multi-parameter measurement in the single-port.

**Figure 4 sensors-23-05191-f004:**
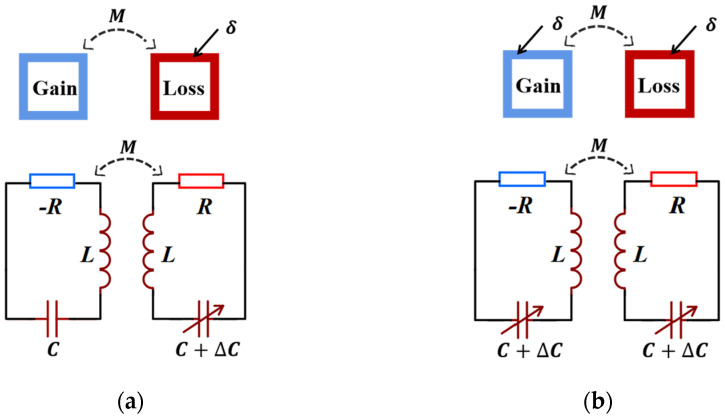
The PT-symmetric-coupled resonator system: (**a**) asymmetric perturbation and (**b**) symmetric perturbation.

**Figure 5 sensors-23-05191-f005:**
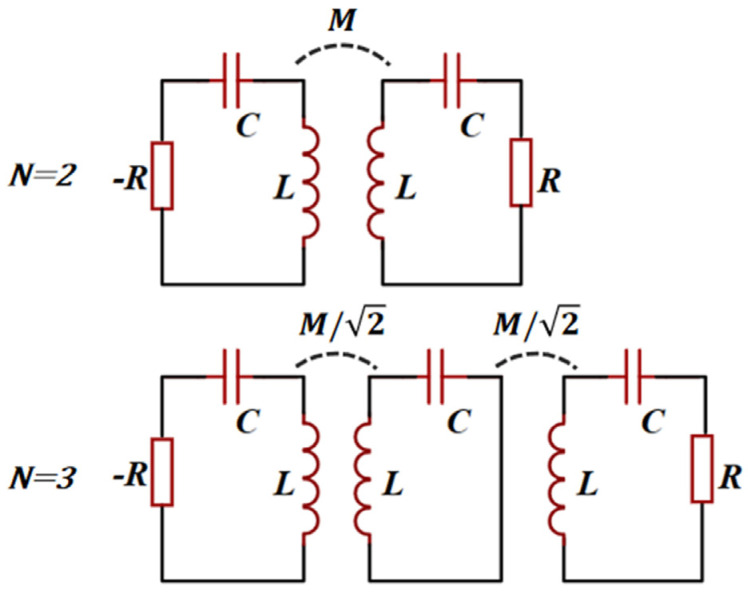
*N*th-order PT-symmetric *RLC* telemetric electric circuit with *N* = 2 and *N* = 3.

**Figure 6 sensors-23-05191-f006:**
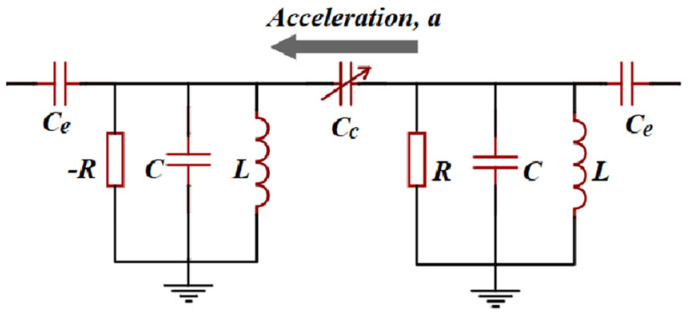
Schematic of the PT-symmetric electromechanical accelerometer.

**Figure 7 sensors-23-05191-f007:**
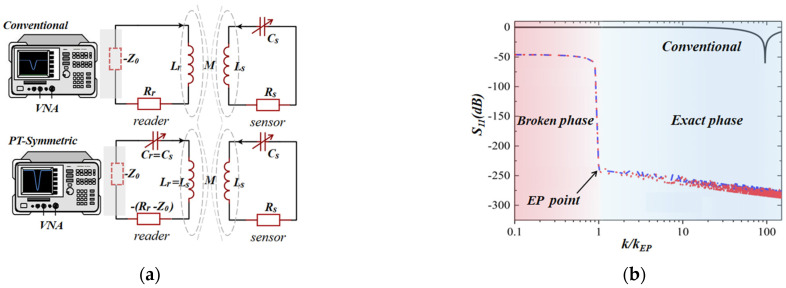
(**a**) The circuit schematic of conventional and PT-symmetric LC passive wireless sensing systems; (**b**) reflection coefficient *S*_11_, at the resonant frequency of the remote system, as a function of coupling strength *k* for conventional and PT-symmetric *LC* passive wireless sensors.

**Table 1 sensors-23-05191-t001:** Application fields of PT symmetry LC passive wireless sensors and numbered references.

Application Fields	References
biomedical and biotelemetry	[50,57,64,65]
industrial measurement and telemetry	[25,36,48,55,56,62,66]
wearable and implantable devices	[49,50,54,57]

## Data Availability

Not applicable.

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
