# Peer review of "PT-Symmetric LC Passive Wireless Sensing"

_sensors, 2023, doi:10.3390/s23115191_

Round 1

Reviewer 1 Report

This manuscript is a review article and, as such, should be written to be accessible to non-specialists. Several changes should be made in order for this manuscript to be a real review:

[1] It is not clear how the PT symmetric sensor enhances read-out distances. Figure 7 is an example where the authors can explain the increased read-out distance, using the plotted functions.

[2] Referring to Figure 7, what are the "conventional sensors"?

[3] Using the sensor for medicine: How does this work? What information does the sensor have and how does the read-out read the information?

[4] There should be a new table where applications in one column and numbered references in another that relate to
  (a) biomedical applications
  (b) bio-telemetry applications
  (c) industrial monitoring
  (d) industrial measurements
  (e) various implantable sensors

[5] The English should be improved in lines 360 and 366.

Minor editing of English in lines 360 and 366.

Reviewer 2 Report

This review provides a comprehensive summary of PT-symmetric LC sensors in various phases, outlining their benefits and debates compared to traditional sensing principles. Despite unresolved issues concerning noise and sensitivity, further research is needed to optimize PT sensor construction.  PT-symmetric LC sensors show promise with enhanced Q-factor, sensitivity, and multi-parameter capabilities, indicating a bright future for LC sensing systems.

This is an interesting work and I recommend its acceptance. Minor comments 

1) The English should be improved 

2) All figures should be replaced by a high quality figures 

Needs some revisions

Reviewer 3 Report

The Authors present a nice and very clear review on status and prospects of PT-symmetric LC sensors in three working areas: exact phase, exceptional point, and broken phase.

In particular, advantages of non-Hermitian sensing concerning classical LC-sensing principles are highlighted.

The work presented meets the criteria of scientific quality, relevance, and robustness for MDPI Sensors.

I recommend the manuscript for publication in its present form.

Reviewer 4 Report

I have carefully read the review sensors-2410991. The Authors present the research status

of PT-symmetric LC sensors in three working areas: exact phase, exceptional point, and broken phase,

demonstrating the advantages of non-Hermitian sensing concerning classical LC sensing principles.

The Authors discussed the benefits of PT circuits, such as higher Q-factor and sensitivity

longer interrogation distance, and multi-parameter sensing method compared to usual circuits.

Also, they point out some open questions as the reduction of noise and the improvement

of the actual sensitivity of the EP sensors. 

Perhaps, the Authors consider including some recent references as Nature Nanotechnology 17(2022)262

and Nature Communication 9(2018)2182. 

I find the manuscript represents an interesting contribution, as a review, in the field of 

non-hermitian dynamics and the application to the design of LC sensors. 

From my previous observations, I consider that the present manuscript should 

be published as a review article in Sensors, after the Authors have 

reviewed the bibliography.

no comments
